# Metabolic Activation of PARP as a SARS-CoV-2 Therapeutic Target—Is It a Bait for the Virus or the Best Deal We Could Ever Make with the Virus? Is AMBICA the Potential Cure?

**DOI:** 10.3390/biom13020374

**Published:** 2023-02-16

**Authors:** Prasanth Puthanveetil

**Affiliations:** Department of Pharmacology, College of Graduate Studies, Midwestern University, Downers Grove, Chicago, IL 60515, USA; pputha@midwestern.edu

**Keywords:** COVID-19, metabolic signaling, PARP, anti-viral therapy, metabolism

## Abstract

The COVID-19 pandemic has had a great impact on global health and is an economic burden. Even with vaccines and anti-viral medications we are still scrambling to get a balance. In this perspective, we have shed light upon an extremely feasible approach by which we can control the SARS-CoV-2 infection and the associated complications, bringing some solace to this ongoing turmoil. We are providing some insights regarding an ideal agent which could prevent SARS-CoV-2 multiplication. If we could identify an agent which is an *activator of metabolism and is also bioactive, we could prevent corona activation* (AMBICA). Some naturally occurring lipid molecules best fit this identity as an agent which has the capacity to replenish our host cells, specifically immune cells, with ATP. It could also act as a source for providing a substrate for host cell PARP family members for MARylation and PARylation processes, leading to manipulation of the viral macro domain function, resulting in curbing the virulence and propagation of SARS-CoV-2. Identification of the right lipid molecule or combination of lipid molecules will fulfill the criteria. This perspective has focused on a unique angle of host-pathogen interaction and will open up a new dimension in treating COVID-19 infection.

## 1. Introduction to Origin of COVID-19

It has been a real mystery tracking the origin of COVID-19 epidemic because the actual cause or origin has not been well defined to date. There have been many assumptions regarding the origin of COVID-19, from bats (Chiroptera species) to pangolins (Manis Pentadactyla species) as their hosts [1], but there are not sufficient data to support any of these claims. One of the earliest commentaries by Wu A et al., 2020, following the outbreak, mentioned the role of 380 amino acid substitutions between severe acute respiratory syndromes (SARS) such as coronavirus (CoV) and the novel SARS-CoV, also termed SARS-CoV-2 [2]. Among these substitutions, the majority of mutations have been reported in the non-structural proteins (nsp) and around 27 amino acid mutations have been reported in the spike proteins, spanning a length of 1273 amino acids [2]. Supporting the claim of pangolin as the intermediate host, it was demonstrated that pangolin CoV and SARS-CoV-2 had over 91% genome homogeneity [3]. The findings also revealed that five amino acids crucial for the binding to human ACE2 receptor were also identical between pangolin CoV and SARS-CoV-2 [3]. Joining all these dots will help us find the right therapeutic agent to combat SARS-CoV-2.

## 2. Development of SARS-CoV-2 Therapeutics

In our frantic search to find an ideal therapeutic for COVID-19, scientists have targeted the virus at different stages of its cycle, starting from cellular entry to intracellular events. Some of the new COVID-19 therapeutics that have generated hope are undergoing drug trials. The available therapeutics mostly target the virus at the host cell entry level or at intracellular sites where the virus undergoes multiplication. Drugs like Chloroquine/Hydrochlorquine, is known to prevent glycosylation of host cell receptors, preventing viral entry and inhibiting the endosomal acidification that could potentially slow down the viral protein synthesis machinery [4]. Even with chemical agents, biologicals and RNA-based therapy in the pipeline, clinical trials with available agents have not demonstrated much promise. Clinical trials involving hydroxychloroquine demonstrated some benefits as a prophylactic against COVID-19 but failed to demonstrate clinical benefits at a statistically significant level based on conclusions [5,6].

Nafamostat and Camostat Mesylate have been shown to inhibit serine protease, TMPRSS2 and finally prevent viral entry [7]. Some of the protease inhibitors used for treating HIV infection, such as Lopinavir/Ritonavir, are also being considered for treating COVID-19 or SARS-CoV-2 [8]. These HIV drugs are expected to have a preventive effect against SARS-CoV-2 multiplication by inhibiting the 3-Chymotrypsin-like protease. The anti-parasitic drug Ivermectin has also been considered as a SARS-CoV-2 therapeutic, making use of its ability to regulate the glutamate-gated chloride channel, thus interfering with the viral entry into the nucleus and preventing viral multiplication [9]. Finally, Remdesivir, a nucleoside analog that has the capability to interfere with viral RNA replication, is also being considered as a potential COVID-19 therapeutic [10,11,12]. With all these top lead therapeutic agents being tried in clinical settings across the globe and many in the pipeline there is potential hope for a cure or stable control of SARS-CoV-2 infection in the near future.

## 3. PARP as a Potential Target and Why Early Activation Is Important

The course of viral entry, expansion/multiplication following invasion of the host cell occurs at different cellular compartments, and the nucleus is the final checkpoint at which we will be able to hinder the virus intrusion and prevent hijacking of the host genetic material. With over 15 family members with known physiological roles [13], present in all organ systems and localized predominantly inside the nucleus [13], PARP family members have recently gained momentum in the area of viral research [14,15,16,17,18]. PARPs’ role has been well studied in the area of DNA damage repair, cell death and host cell defense [13,14,17,19], but their role in the area of viral research is a bit controversial because they exhibit properties both for and against the viral invasion.

It was demonstrated that PARP1 inhibition during adenoviral infection promotes E4orf4 enzyme activity, resulting in downregulation of DNA damage response signaling and PARP-mediated ADP ribosylation (PARylation) modification and finally promoting viral replication [20,21]. PARP1 present in host cells is a critical factor for the proper function of human, avian and swine Influenza A virus RNA-dependent RNA polymerase activity based on studies in human 293T cells [22]. It was demonstrated in human lung A549 cells that PARP1 is a crucial factor for effective viral nucleoprotein formation [22]. In the same study, findings have revealed that when PARP-mediated PARylation was inhibited, viral RNA-dependent RNA polymerase activity was enhanced [22]. This study emphasizes the necessity of a PARP1 protein presence for both viral RNA-dependent RNA polymerase activity and nucleoproteins but not PARylation modification [22]. In fact, enhanced PARylation of viral proteins could inhibit viral RNA-dependent RNA polymerase activity. Ideally, at the moment when the virus comes into contact with the host cell PARP protein, if we can activate PARP1-mediated PARylation of viral proteins, we could inhibit viral replication. Regarding corona virus family members, the role for PARP family members is yet to be discussed. Grunewald ME et al., 2018, demonstrated that ADP ribosylation of viral nucleocapsid protein is one of the major post-translational modifications that occurs during a viral invasion or infection, a process facilitated by PARP family members [23]. The study demonstrates that this crucial modification of the nucleocapsid occurs in porcine epidemic diarrhea virus and also in corona virus family members, including Severe Acute Respiratory Syndrome—Corona Virus (SARS-CoV), and Middle East Respiratory Syndrome—Corona Virus family members (MERs-CoV) [23]. Even though the nucleocapsid protein undergoes PARylation modification in Corona Virus family members, how this modifies Corona virus function, specifically SARS-CoV-2 virulence and ability to undergo multiplication, is yet to be understood. In this area of research some clarity was brought by the work of Li C et al., 2016, in which the authors suggested that viral macrodomains had the ability to inhibit or reverse the ADP ribosylation processes due to an enzyme hydrolase property, resulting in de-MARylation and de-PARylation [24]. This property was demonstrated by macrodomains of not only Hepatitis E virus (HEV), but also Venezuelan equine encephalitis virus (VEEV) and Severe acute respiratory syndrome coronavirus (SARS-CoV) [24]. In this work, the authors demonstrated that SARS-CoV and VEEV macrodomains had a better ability to reverse the ADP ribosylation, especially the de-MARylation compared to HEV [24]. Considering this observation, de-MARylation by macrodomain of SARS-CoV could be playing a major role in enhancing its replication and virulence. This work was followed by another interesting study by Grunewald ME et al., 2019, in which the authors confirmed that coronaviridae family members require macrodomains to prevent PARP protein-mediated inhibition of viral replication and also to prevent a PARP-mediated interferon response [18]. Viral macrodomains, as seen with alpha viruses and corona viruses, have been demonstrated to have poly(ADP-ribose) hydroxylase activities which in turn could aid in their own multiplication [25]. In a review by Leung AKL et al., 2018, it is explained clearly how viral macrodomains could try to hinder the anti-viral defense by PARP-mediated mechanisms, by enhancing (ADP-ribosyl)hydroxylase/hydrolase activity [26]. There are also some studies which have detailed the beneficial effects of PARP-1 inhibition. A flavonol family member, morin, has been demonstrated to have anti-viral activity by targeting PARP1 in vitro [27]. Another in vitro study, using stenoparib, which is a knownPARP1 inhibitor, demonstrated a dose-dependent inhibition of SARS-CoV-2 variants [28]. Interestingly, there are no in vivo studies or human studies involving morin or stenoparib to further corroborate the in vitro findings. Based on some of these few crucial findings, PARP12, PARP14 and PARP15 established their role as the major PARylating enzymes targeting viral macrodomains and have got the capacity to restrict viral replication following ADP-mediated ribosylation modification of the viral macro domains [18]. It was also demonstrated that pan-PARP inhibition and PARP 12 and 14 silencing could lead to increased viral replication, especially in a mutant virus [18]. This study also revealed that PARP 14 was also involved in the induction of interferon synthesis, an additional mechanism that could contribute towards antiviral defense [18]. These findings help us understand why it is very crucial to activate PARPs during the early stages of viral replication and to prevent invasion of host cells.

## 4. The Debate—Is PARP Activation a Bait for SARS-CoV-2 or the Best Deal We Could Ever Have?

It is very important to have this debate because of the complexity of the interaction between SARS-CoV family members and the host cell PARP protein family members. The presence of PARP protein family members is required for effective viral replication [22] (inert or uncleaved PARP) but PARP activation could lead to inhibition of viral replication [18,29]. Activation of PARP could lead to enhanced ADP ribosylation of viral macrodomains, either MARylation or PARylation, which contributes to the decreased virulence and reduced replication ability of the virus [18,29]. From a therapeutic perspective, if we inhibit PARP family members, we are indirectly contributing towards a decreased ADP ribosylation of viral macrodomains, resulting in hindering our own innate defense ability against the virus. In the phase when the viral count is high or when the host immune system is weak, the virus has the upper hold over the host cell. At this point, the virus tries to reverse the ADP ribosylation processes and make use of PARP proteins for the viral nucleoprotein formation and thus enhance viral replication. At the same time, the host cell could win this game with the virus if the host cell could activate the existing PARP protein family members, thus enhancing their host cell’s ADP ribosylation ability [18,29]. Activation of host cell PARP activity at this juncture could then lead to an enhanced MARylation and PARylation of viral macrodomains, inhibiting their virulence and replication [18,29].

## 5. Determining Factors for ADP Ribosylation of Viral Macrodomains

For an effective ribosylation modification to occur we need a sufficient amount of specific substrates and co-factors that could facilitate this process. These include substrates such as ATP, ADP and the co-factor NAD + [29,30]. One of the important features of any viral infection is the virus’s ability to deplete the host cell ATP levels. In the presence of a depleted ATP state, there is a gradual decline in the levels of nucleotide breakdown products, ADP and AMP. The PARP-dependent ribosylation reaction directly depends on available ADP levels [31]. During the phase of viral invasion of host cells, when the ATP levels drop, we may have enough substrate in the form of ADP during an early stage of infection but even ADP levels depletes shortly [31]. As a result of ADP depletion, the ADP ribosylation modifications such as MARylation and PARylation cannot go to exhaustion and ultimately this could diminish the host’s antiviral defense. Some of the virus family members have the machinery to abolish the MARylation process occurring at the macrodomain. Some of the reports have demonstrated that Corona virus family members and members belonging to the single-stranded RNA superfamily, including alphaviruses such as the Chikungunya virus, non-alpha viruses such as the Hepatitis E virus, and matonaviridae family members such as the rubella virus, all have the ability for de-MARylation. The SARS-CoV-2 macrodomain has also been shown to have the ability to de-MARylate by its (ADP-ribosyl)hydroxylase activity [32,33]. This continuous conflict between host-cell-mediated viral macrodomain MARylation and PARylation and viral macrodomain-mediated de-MARylation due to (ADP-ribosyl)hydroxylase activity is a major determining factor. The more the host cell could combat the viral invasion, the higher the chances for effective ADP ribosylation modification of the viral macrodomain; this state is also dependent on the available ATP for the host immune and non-immune cells to perform this defensive action against virus.

## 6. How Can WE Make a Deal with SARS-CoV-2?

Following an infection due to viral invasion, the host cell is already undergoing ATP crisis and during that interval, activation of PARP family members could lead to further depletion of available ATP reserves. This ultimately leads to the host cell getting stranded without having enough substrate, i.e., the ADP required for effective ribosylation modification. PARP activation as a sole therapeutic strategy without giving much thought to how to provide the host cell with necessary substrates such as ATP and ADP will be just a deal with the virus that comes with pros and cons. The pros would be that the sooner we are able to activate PARP members and prevent viral replication, the sooner we will be able to stop infection advancing to its critical stage. The cons would be that our ability to restrict viral replication will be possible only until there is a sufficient ADP reserve inside the host cell. Under these above-mentioned conditions, the sooner we are able to activate PARP with the direct provision of ATP or ADP, the sooner we will be able to limit the disease. The availability of host cell ADP and ATP depends upon multiple determining factors, including pathogenic virulence, host cell health and the ability of a host cell to adapt to the metabolic shift following a viral invasion [31,34]. If we decide to metabolically activate PARP, we will be able to handle the crisis. The agent should be an Activator of Metabolism, Bioactive and Inhibitor of Corona Activation (**AMBICA**). Ideally, we should select a metabolic substrate, which is bioactive and which is capable of activating cellular metabolism and generating ATP for host cells that could result in both PARP activation and ATP provision of host cells.

## 7. How to Convert the Deal into a Bait for SARS-CoV-2?

We will be able to convert this deal into a bait for virus only if the host cell has an upper hold, and this will occur only if we provide the host cell with sufficient energy sources to combat the ongoing viral invasion and expansion. Crucial findings by Ivanov K A et al., 2004, and Hoffman M et al., 2006, using molecular biology tools and computer-aided design techniques have demonstrated and confirmed that ATPase of SARS CoV Helicase catalytic subunit binds tightly to substrate ATP [35,36]. The complex formed between viral ATPase and host cell ATP is a most stable complex, ATP being the most preferred substrate for binding of viral ATPase of any other substrate [35,36]. These studies also reported that ATP binding to the ATPase subunit of SARS CoV Helicase is a stronger and more stable complex than binding to any ribose or deoxyriboses as substrate [35,36]. It is possible that the binding of ATP by viral ATPases of the helicase complex could be releasing ADP. Following ATP breakdown to ADP, they could act as substrates for PARP-mediated ADP ribosylations, the MARylation and PARylation in the specific cellular location. The next question would be: how will we activate PARP and provide ATP simultaneously to the host cell without depleting much of the host cell ATP reserve? Moreover, what agent is best suited to providing this dual effect?

## 8. The Need for Metabolic Activation of PARP—Considering the ATP Requirements

An ideal strategy would be to use an agent that can activate PARP and provide ATP simultaneously. One of the options would be to adopt a chemical or biological agent to target PARP that could enhance PARP expression and/or activity’ however, this strategy is not applicable over the long term. In the short term it could lead to a sudden depletion of the available host ADP, due to triggering of ADP ribosylation modification. The next option would be to give a combination of PARP activator and ATP exogenously. In this case, we will not be able to get the desired effect because exogenously provided ATP may produce nonspecific effects acting on purinergic receptors throughout the body [37], bringing about unanticipated positive and negative effects. The agent should be able to generate ATP in the host’s intracellular compartment, along with its ability to activate PARP family members. Ideally, the agent should be an activator of metabolism and at the same time bioactive and readily available as we introduce it into the host’s body, and it should possess the capability of inhibiting corona activation or aggression.

## 9. PARP, NAD+ and Changes in Cellular Metabolome and its Impact on COVID-19 Infection

The PARP family members, specifically PARP1, PARP2, PARP7, PARP10, and PARP 11, have been demonstrated to have metabolic roles [38,39,40]. PARP family members have the ability to alter the core of the metabolome of a cell due to their influence on NAD+ co-factors. NAD+ is a major co-factor involved in diverse biochemical pathways [41,42,43,44,45,46,47,48]. Along with aiding biochemical pathways involved in ATP generation, NAD+ also acts as a co-factor for PARP family members and histone deacetylases, such as Sirtuin family members (Sirtuin 1–7) [41,42,43,44,45,46,47,48]. PARP1 activity is dependent on NAD+ availability [41,42,43,44,45,46,47,48]. A NAD+ mediated increase in PARP activity leads to an enhanced ADP ribosylation process and an increase in Sirtuin activity leads to histone de-acetylase activity, contributing towards epigenetic changes [41,42,43,44,45,46,47,48,49,50,51,52]. Due to this competing nature of PARP and Sirtuin family members for NAD+, it is mandatory to keep the PARP levels intact [41,42,43,44,45,46,47,48]. In the absence of PARP availability, NAD+ will be utilized by the Sirtuin family members [41,45,46,49,50,51,52]. The amount of NAD+ is also tightly controlled within the cell [41,45,46,49,50,51,52]. Once the NAD+ undergoes depletion, even in the presence of an enhanced PARP expression, PARP family members will not be able to perform their function, i.e., ADP ribosylation [41,53]. This can be considered as a negative feedback loop for limiting unregulated PARP activity [41,53]. This also validates our claim that metabolic activation of PARP is highly important. Metabolic activation leads to the ATP generation required to energize host cells, enhancing NAD+ and augmenting PARP-mediated ADP-ribosylation activity. Once substantial ATP is generated inside the host cells, NAD+ levels drop and subsequently PARP activity (ADP-ribosylation) falls. This phenomenon could also potentially explain why NAD+ boosters could be a failure for curbing SARS-CoV-2/COVID-19 infections. The commercially available NAD+ boosters, such as Nicotinamide mononucleotide (NMN), Nicotinamide riboside (NR), and Nicotinamide adenine dinucleotide hydride, are capable of raising cellular NAD+ levels, but they come with potential drawbacks [54,55]. Overactivation of NAD+ could result in activating PARP and Sirtuin family members in an unregulated manner [41,45]. An unregulated PARP activation in the absence of a negative feedback loop could turn on necrotic signaling that could potentiate the COVID-19-mediated deteriorating effect on the body. Providing extracellular NAD+ could turn on the body’s ability to activate PARP family members. As a result, it could deplete the ATP reserves of the host cell, specifically immune cells, making them unable to combat the viral attack. Along with being a metabolic activator, it is also mandatory to have an agent which is bioactive, at the same time safe and tolerable for the cellular environment. An agent capable of activating metabolism and simultaneously bioactive will be capable of providing ATP and triggering the PARP-NAD+ nexus, leading to MARylation and PARylation of viral macrodomains, curbing the SARS Cov infection and their multiplication inside host cells.

## 10. PARP, COVID-19 and Clinical Trials

It is quite intriguing that none of the PARP regulators even reached the clinical trials. Research groups working on both PARP activators or PARP inhibitors have an equal chance of failing due to the lack of insight regarding how and when PARP should be regulated. The clinical trial using Stenoparib, an orally administered PARP inhibitor, is expected to demonstrate anti-viral activity against SARS-CoV-2 and its variants [28,56,57]. A follow-up in vitro study published recently used dose-dependent effects of Stenoparib on Vero E6 monkey kidney cells and Calu-3 human lung adenocarcinoma cells against a SARS-CoV invasion [56]. Even though the results demonstrated a positive effect on curbing viral infection in cells, the effects were due to a combination with remdesivir [56]. The ability of Stenoparib to curb the virus per se both in vitro or in the whole animal model or in clinical subjects is not proven. The potential long-term impact of Stenoparib administration on systemic health will be another challenge for subjects recovering from COVID-19 infection. It would be ideal to test an agent that is a metabolic activator, bioactive, therapeutically effective and tolerable to human cells, which is lacking at this point.

## 11. Discussion and Conclusions

Which agent best fits the description of an Activator of Metabolism, Bioactive and Inhibitor of Corona Activation (**AMBICA**)? Gariani K et al., 2017, show along a model of NASH, and ASH, also in high-fat, high-sucrose-diet-fed mice, that there is an increase in PARP1/12/14 expression, cleavage and increase in PARylation [58]. The diet has 61% of saturated fatty acids among the fat content [58]. Moraes J C et al., 2009, demonstrated that feeding a high-fat diet that contained saturated fats as the major fatty acid components was able to elevate the expression of PARP in hypothalamic neurons [59]. Scheibye-Knudsen M et al., 2014, also demonstrated that a high-fat diet was able to activate PARP and PAR formation in mice [60]. The diet fed in this study also had 60% saturated fat [60]. The correlation is made on diet-based studies because there are only limited studies that have focused on the relationship between dietary fat composition and SARS-CoV-2 infection. As the COVID-19-related research expanded, angiotensin-converting enzyme 2 (ACE2) emerged as an anchoring target on cells aiding cellular entry for SARS-CoV-2. Graus-Nunes F et al., 2019, demonstrated a negative correlation between a high-fat diet and ACE2 expression [61]. Findings have also demonstrated this negative correlation and provided more information on the diet, containing 58% high-fat and mostly comprising of coconut oil, which is a lauric acid(in major proportion) and palmitate (in minor proportion) containing source [62]. Now the major question we need to ask is: which among the known saturated fatty acids—lauric, myristic, palmitic and stearic will be able to fulfill the criteria? Or is it a specific combination of these saturated fatty acids which could serve as Activator of Metabolism, Bioactive and Inhibitor of Corona Activation (AMBICA)? which could facilitate the process of ADP Ribosylation of viral macrodomains along with providing sufficient amount of ATP for the host immune cells (Figure 1).

Once we figure out what ideal concentrations of fatty acids represent what constitutes AMBICA, we could set the bait for SARS-CoV-2, and this will be an authentic agent to curb COVID-19 infection. This could really help in tackling the COVID-19-associated health complications globally along with the approved agents currently in use.

## 12. Limitations

One of the limitations or challenges is the perception regarding the use of a PARP activator. If the actual grasp of the context is lost, then we will be scrambling in the same plane. It is very important to understand that PARP-mediated MARylation and PARylation of viral macrodomains should occur at a very early stage of viral invasion or infection. The stage of viral infection is quite a determining factor. If we tend to provide AMBICA (Activator of Metabolism, Bioactive and Inhibitor of Corona Activation) at later stages of infection, we will not get the desired beneficial effects. What is the right combination of fatty acid components that constitute AMBICA are totally unknown at this stage. The positive side is that we know about these individual saturated fatty acid components which are well studied in animal models and humans, but at this stage we do not have sufficient experimental evidence to claim whether a single saturated fatty acid or a specific combination of saturated fatty acids would fulfill the criteria for AMBICA. Considering the availability and accessibility for these agents (fatty acids) for most researchers, a prompt investigation into identifying the composition and its impact on SARS-CoV-2 will reveal crucial information and is definitely mandated.

## Figures and Tables

**Figure 1 biomolecules-13-00374-f001:**
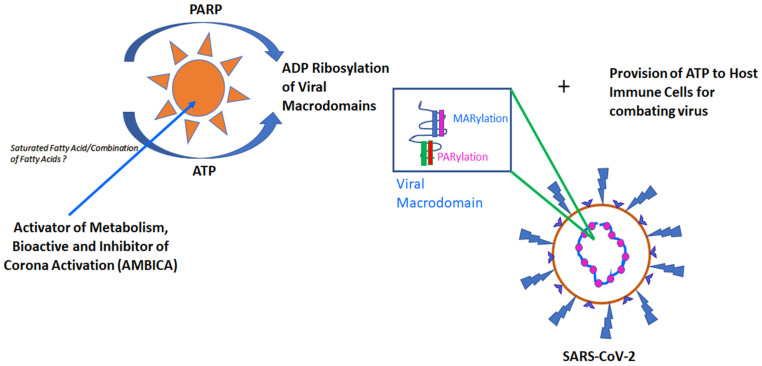
An ideal agent should be an activator of metabolism and bioactive with the capability of inhibiting corona activation (AMBICA). A specific saturated fatty acid or combination of same in specific proportion could be the best fit to act as AMBICA. This agent should be able to provide ATP for cellular fueling, especially for host immune cells, and provide the initial breakdown product of ATP, i.e., ADP as substrate for PARP-mediated MARylation and PARylation of the SARS-CoV macrodomain. This simultaneous property of providing a host cell with ATP as a fuel source and ADP for PARP-mediated MARylation and PARylation will help in preventing SARS-CoV multiplication without the host cells getting deprived of ATP.

## Data Availability

Not applicable.

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
