# Peer review of "Metabolic Activation of PARP as a SARS-CoV-2 Therapeutic Target—Is It a Bait for the Virus or the Best Deal We Could Ever Make with the Virus? Is AMBICA the Potential Cure?"

_biomolecules, 2023, doi:10.3390/biom13020374_

Round 1
Reviewer 1 Report
This is an interesting paper about a new metabolic perspective in dealing with COVID-19 infection. I have some comentaries and sugestions about this paper:
1. In the section about therapeutics, maybe you can add some clinical trials to support that findings. There are some new trials in 2022 that present new information about hydroxychloroquine (https://pubmed.ncbi.nlm.nih.gov/35139097/ ; https://pubmed.ncbi.nlm.nih.gov/35943669/
https://pubmed.ncbi.nlm.nih.gov/34326192/
2. In the section about PARP as a potential target, are cited many studies about PARP role; the findings are explained and corelated; also, I suggest to add another new studies, clinical trials more recently published; there are some new studies about some inhibitors used in ACE2 target and PARP inhibition that could block SARS-Cov2 replication:
https://pubmed.ncbi.nlm.nih.gov/33526003/
https://pubmed.ncbi.nlm.nih.gov/36103462/
https://pubmed.ncbi.nlm.nih.gov/35297669/
3. I suggest to add a discussion section and/or a conclusion section.
4. I suggest to make an updated on references, maybe you can find more references recently published.
Author Response
Reviewer - 1
This is an interesting paper about a new metabolic perspective in dealing with COVID-19 infection. I have some commentaries and suggestions about this paper:
Response - We sincerely thank our reviewer. The responses are addressed below. The changes in the revised manuscript for the reviewer -1 is marked as Blue.
- In the section about therapeutics, maybe you can add some clinical trials to support that findings. There are some new trials in 2022 that present new information about hydroxychloroquine
https://pubmed.ncbi.nlm.nih.gov/35139097/ ;
https://pubmed.ncbi.nlm.nih.gov/35943669/
https://pubmed.ncbi.nlm.nih.gov/34326192/
Response: We sincerely thank the reviewer for the valuable input. We have now added the suggest references and added information included in the clinical trials to our revised version of the manuscript.
- In the section about PARP as a potential target, are cited many studies about PARP role; the findings are explained and corelated; also, I suggest to add another new studies, clinical trials more recently published; there are some new studies about some inhibitors used in ACE2 target and PARP inhibition that could block SARS-Cov2 replication:
https://pubmed.ncbi.nlm.nih.gov/33526003/
https://pubmed.ncbi.nlm.nih.gov/36103462/
https://pubmed.ncbi.nlm.nih.gov/35297669/
Response: Once again we appreciate the reviewer for providing valuable critique. We will explain the paragraph in the revised version based on the comments provided by the reviewer.
- I suggest to add a discussion section and/or a conclusion section.
Response: We have now made changes as suggested by our reviewer. The revised manuscript has been provided with a discussion and conclusion section.
- I suggest making an updated on references, maybe you can find more references recently published.
Response to reviewers: We agree with the reviewer and have added some of the newer references.
Reviewer 2 Report
This review/opinion article on the suitability of targeting PARPs as a therapeutic agent for SarsCOV2 infections was very interesting. The author has clearly thought in detail about the pros-cons of PARP inhibition/activation and the role of energy metabolism in the inter-relationship between PARP activation and effective pathogen control. The author has carefully synthesized available literature on this subject and interjected his own ideas to suggest options for the probable ways to target PARPs as therapeutics, making this article thought-provoking and of considerable interest to people working in infectious disease research. Some minor comments to address:
1. Line 34-35 pls check grammar-rewrite.
2. Line 78 mentions when Parp1 parylation is decreased viral polymerase activity is enhanced but Line 83 mentions as Parp parylation is enhanced viral polymerase is enhanced, pls check and clarify/correct.
3. Throughout the manuscript the intext citations seem to include author data full information on journal article, page number etc. Usually, the author-date format is sufficient. Please format in-text citations to be of uniform format.
Author Response
This review/opinion article on the suitability of targeting PARPs as a therapeutic agent for SarsCOV2 infections was very interesting. The author has clearly thought in detail about the pros-cons of PARP inhibition/activation and the role of energy metabolism in the inter-relationship between PARP activation and effective pathogen control. The author has carefully synthesized available literature on this subject and interjected his own ideas to suggest options for the probable ways to target PARPs as therapeutics, making this article thought-provoking and of considerable interest to people working in infectious disease research. Some minor comments to address:
Response: We sincerely thank the reviewer for understanding the relevance of the topic we addressed in the manuscript and mentioning it thought provoking. We have made a sincere attempt to answer all the concerns. The responses are marked in Red in the revised version.
- Line 34-35 pls check grammar-rewrite.
Response – We thank the reviewer for correcting us and have checked the grammar errors and corrected in the lines 34-35 as suggested.
- Line 78 mentions when Parp1 parylation is decreased viral polymerase activity is enhanced but Line 83 mentions as Parp parylation is enhanced viral polymerase is enhanced, pls check and clarify/correct.
Response – We apologize for the confusion. Just would like to clarify that as mentioned in line 78, PARP Parylation is inversely related to viral polymerase activity. The sentences between 78- 83 now reads as below and it indicates that when PARP1 mediated PARylation of viral proteins we could prevent viral replication. Just to bring in more clarity, the virus requires PARP proteins for it’s replication, but not an active protein where PARP1 possesses PARylating ability. With enhanced PARylating ability we could prevent the viral multiplication.
“Infact, enhanced PARylation of viral proteins could inhibit viral RNA dependent RNA polymerase activity. Ideally the moment when virus comes in contact with the host cell PARP protein, if we can activate PARP1 mediated PARylation of viral proteins, we could inhibit viral replication.”
- Throughout the manuscript the intext citations seem to include author data full information on journal article, page number etc. Usually, the author-date format is sufficient. Please format in-text citations to be of uniform format.
Response – We sincerely thank the reviewer for the comment. The formatting regarding the intext citations is now corrected as per reviewer’s suggestion in our revised manuscript.
Round 2
Reviewer 1 Report
Dear authors,
Thank you for addressing all the suggestions I made. I thing that you improved the quality of this manuscript.